# Impact of transport electrification on critical metal sustainability with a focus on the heavy-duty segment

Han Hao [1,2,3], Yong Geng[4,5,6]*, James E. Tate [7], Feiqi Liu[1,3], Kangda Chen[1,3], Xin Sun[1,2], Zongwei Liu[1,3] & Fuquan Zhao[1,3]

The majority of transport electrification studies, examining the demand and sustainability of critical metals, have focused on light-duty vehicles. Heavy-duty vehicles have often been excluded from the research scope due to their smaller vehicle stock and slower pace of electrification. This study fills this research gap by evaluating the lithium resource impacts from electrification of the heavy-duty segment at the global level. Our results show that a mass electrification of the heavy-duty segment on top of the light-duty segment would substantially increase the lithium demand and impose further strain on the global lithium supply. The significant impact is attributed to the large single-vehicle battery capacity required by heavy-duty vehicles and the expected battery replacement needed within the lifetime of heavy-duty vehicles. We suggest that the ambition of mass electrification in the heavy-duty segment should be treated with cautions for both policy makers and entrepreneurs.

[1] State Key Laboratory of Automotive Safety and Energy, Tsinghua University, 100084 Beijing, China. [2] China Automotive Energy Research Center, Tsinghua University, 100084 Beijing, China. [3] Tsinghua Automotive Strategy Research Institute, Tsinghua University, 100084 Beijing, China. [4] School of International and Public Affairs, Shanghai Jiao Tong University, 200240 Shanghai, China. [5] Shanghai Institute of Pollution Control and Ecological Security, 200092 Shanghai, China. [6] China Institute for Urban Governance, Shanghai Jiao Tong University, 200030 Shanghai, China. [7] Institute for Transport Studies, University of Leeds, Leeds LS9 2JT, UK. *email: ygeng@sjtu.edu.cn

Electrification of the transport sector is an essential measure to cope with global energy and climate challenges. Plug-in electric vehicles (PEVs), including battery electric vehicles and plug-in hybrid electric vehicles, have made remarkable progress in the light-duty vehicle (LDV) segment, namely, passenger vehicles and light-duty commercial vehicles. In 2018, global electric car sales reached 2.08 million, accounting for 2.2% of total car sales[1]. In countries leading the electrification of their vehicle fleets, such as Norway, the PEV share of new car sales exceeded 40%[1]. In comparison, the progress in electrifying the heavy-duty vehicle (HDV) segment is slower. HDVs comprising heavy-duty trucks and buses account for only 10% of the global vehicle stock but are responsible for 46% of the greenhouse gas emissions from road transport[2]. Electrifying the HDV segment can lead to substantial energy, climate and air quality benefits[3]. Intensive efforts are being made by global entrepreneurs to achieve breakthroughs in HDV electrification technologies and production[2]. For example, the prominent electric vehicle manufacturer Tesla is seeking to extend its successful experience in LDV electrification to the HDV segment by launching the Tesla Semi[4], a heavy-duty electric truck with an electric range of 800 km. The Tesla Semi, as well as a number of similar electric HDV models being developed by other vehicle manufacturers, are intended to phase out the conventional dominance of the commercial transport industry by diesel trucks.

Compared with conventional HDVs, electric HDVs have both advantages and disadvantages. The most significant advantage is the reduction of energy cost. As reported by Tesla, the estimated energy cost to operate a Tesla Semi is only half that of a conventional diesel truck, implying a 2-year payback period[4]. Such a cost advantage would undoubtedly shock this market with operators that are highly sensitive to investment and operating costs[5]. In addition, HDVs often operate on fixed routes, such as delivery vehicles and urban buses, which could be effectively covered by a relatively small number of location-specific charging stations. Unfortunately, a key challenge in this segment is the high energy demand to propel large vehicles and their loads. This challenge is further exacerbated for long-distance coaches and tractor trailers that have large driving ranges. The level of battery capacity needed by HDVs is significantly higher than that required by LDVs. The expected large battery capacity puts burdens on both vehicle weight and cost. Furthermore, the lifespan of HDVs is significantly longer than the lifespan of batteries. The average accumulated mileage of heavy-duty tractor trailers can be greater than 1,200,000 km[6–8]. Battery replacement is expected to be required during a HDV's lifetime, further increasing demand for batteries and their raw materials, such as lithium, cobalt, and nickel.

Previous studies have intensively investigated the resource impacts from LDV electrification and uncovered considerable resource challenges (see summary in Supplementary Table 1). In contrast, the resource impacts from HDV electrification have received scant attention. Here we evaluate the resource impacts from mass electrification in the HDV segment by using lithium as one example. A technology-rich, bottom-up approached model is established to simulate the resource impacts. The results show that mass electrification of the heavy-duty segment would significantly increase lithium demand and should be treated with cautions.

## Results

**Future trends**. The results are presented based on four established scenarios (D1, D2, D3, and D4) distinguished by market penetration of PEVs, vehicle electric range, and battery durability (see the Methods for details). Figure 1 shows the annual lithium inflow (defined as gross demand), outflow and stock associated with global vehicle fleet by using scenario D2 as one example (under which both the LDV and HDV segments are electrified). The results for the other scenarios can be found in Supplementary Figs. 1, 2, 3. Under scenario D2, the lithium gross demand is projected to reach 0.65 mt by 2050 and 2.63 mt by 2100. Out of the total gross demand in 2100, the contributions from LDV manufacturing, LDV battery replacement, HDV manufacturing, and HDV battery replacement are 1.31 mt (50%), 0 mt (0%), 0.72 mt (27%), and 0.60 mt (23%), respectively. The lithium gross demand from HDVs will be at a similar level with that from LDVs. The contribution from HDV battery replacement is significant, accounting for approximately one quarter of the total gross demand. As the volumes of vehicles and batteries increase, the lithium outflow grows. The total lithium outflow will reach 0.30 mt by 2050 and 2.22 mt by 2100. In terms of lithium stock, 27.5 mt of lithium will be stocked in the global vehicle fleet by 2100, out of which 19.5 mt (71%) will be in the LDV segment and 8.0 mt (29%) will be in the HDV segment.

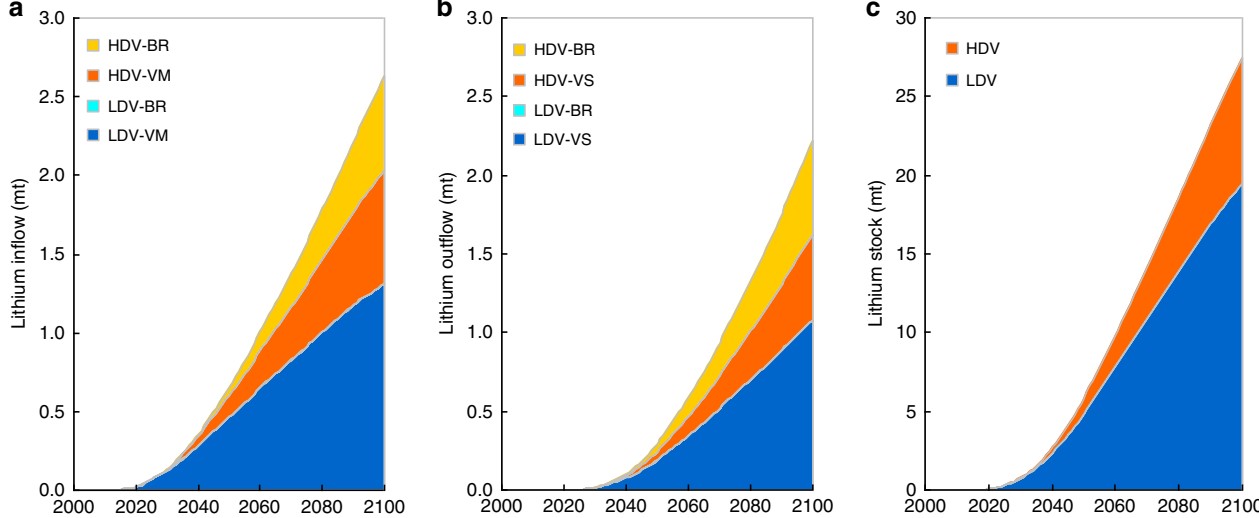

**Fig. 1** Annual lithium inflow to, outflow from and stock in global vehicle fleet. The subfigures show the annual lithium inflow (**a**), outflow (**b**), and stock (**c**). The results are based on scenario D2. VM: Vehicle Manufacturing; VS: Vehicle Scrappage; BR: Battery Replacement. Source data are provided as a Source Data file.

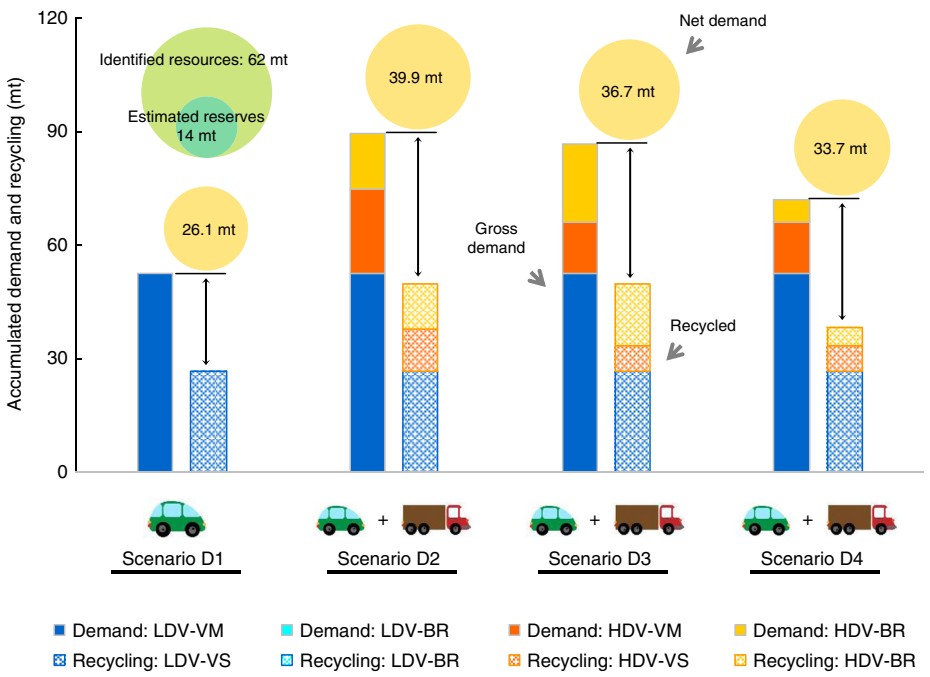

**Fig. 2** Accumulated lithium demand and recycled lithium under different scenarios. The four groups of bars represent the results under scenarios D1, D2, D3, and D4, respectively. For each group of bars, the left bar indicates the accumulated gross demand; the right bar indicates the accumulated recycled lithium; the difference in the length of the two bars indicates the accumulated net demand, which is also indicated by the area of the circles above the bars and the numbers within the circles. The green circles represent the current estimates of global lithium reserves and the total identified lithium resources for comparison with the accumulated net demand. VM: Vehicle Manufacturing; VS: Vehicle Scrappage; BR: Battery Replacement. Source data are provided as a Source Data file.

**Accumulated resource impacts**. Figure 2 shows the 2000–2100 accumulated lithium gross demand, recycled lithium and net demand (the part of gross demand that is met by primary resources) under different scenarios (see also Supplementary Table 2). With vehicle electrification restricted in the LDV segment (scenario D1), the accumulated gross demand is estimated to be 52.6 mt. All the demand is from vehicle manufacturing, with no contribution from battery replacement. With the assumed end-of-life recycling rate, 26.5 mt of lithium can be recycled as secondary supply, implying an accumulated net demand of 26.1 mt. With resource efficiency of the production chain considered (90% for lithium-ion battery production), such a net demand is equivalent to 29.0 mt of lithium mine production, which is higher than the current estimates of global lithium reserves (14 mt) and accounts for almost a half of total identified lithium resources (62 mt)[9]. It should be noted that the resources and reserves estimations are dynamic. Future lithium reserves and resources estimates are expected to increase with growing demands. The static resources and reserves estimates adopted by this study, based on the current technological and economic conditions, are used as benchmarks to highlight the magnitude of the resource challenge.

If the HDV segment becomes further electrified (scenario D2), 36.9 mt of lithium gross demand is added by HDVs, inducing the total gross demand up to 89.5 mt, 70% greater than the level under scenario D1. Of the 36.9 mt incremental lithium gross demand by the HDV segment, 22.2 mt (60%) is from vehicle manufacturing, and 14.8 mt (40%) is from battery replacement. Subtracting the 49.6 mt lithium that could be recycled, the net demand under scenario D2 is 39.9 mt, 53% higher than the level under scenario D1. This net demand increase caused by HDV electrification substantially widens the difference between the net

demand and supply capacity. Under such a circumstance, a great pressure on the global lithium supply is expected.

With an assumed reduction in the HDV electric range (scenario D3), the lithium gross demand from HDVs changes significantly. On the one hand, the lithium gross demand from HDV manufacturing decreases from 22.2 mt to 13.3 mt. This decrease reflects the impact from reduced single-vehicle battery capacity. On the other hand, the lithium gross demand from HDV battery replacement increases from 14.8 mt to 20.6 mt. The underlying reason behind this increase is that reducing the electric range of HDVs causes a reduction in the km-measured battery lifespan (obtained by multiplying the battery cycle life and vehicle electric range) and eventually induces more frequent battery replacements. Overall, scenario D3 is estimated to lead to 86.5 mt of gross demand and 36.7 mt of net demand, 3% and 8% lower than the values under scenario D2, respectively.

If battery durability is further improved (scenario D4), the lithium gross demand from battery replacement in HDVs can be effectively reduced, from 20.6 mt under scenario D3 to 6.0 mt under scenario D4. The corresponding total gross demand and net demand would decrease to 71.8 mt and 33.7 mt, 17% and 8% lower than the values under scenario D3, respectively. Although the lithium demand under scenario D4 is considerably reduced, the net demand is still 29% greater than that under scenario D1, implying a robust challenge from HDV electrification.

It should be noted that the estimations do not cover lithium demand from sectors other than PEVs. Existing studies show that lithium demand from other sectors is expected to grow in a relatively mild pattern in the coming decades. With an assumed annual growth rate of 5%, the gross demand is expected to reach around 0.2 mt by 2050[10], equivalent to ~30% of the expected lithium gross demand from PEVs (under scenario D2). This

modest but not insignificant additional demand adds further resource pressures on HDV electrification.

## Discussion

The results suggest that global lithium resources will not be able to sustain simultaneous mass electrification of both the LDV and HDV segments. Since the electrification in the LDV segment has already imposed significant strains on the global lithium supply, further mass electrification in the HDV segment, which is expected to increase the accumulated net demand by 29% to 53%, would come with risks. Even if electric HDVs gain a techno-economic advantage over other powertrain technologies and achieve market success in the short term, their long-term development is likely to face resource constraints with a reflected surge in lithium prices. It is therefore recommended that both the government and vehicle manufacturers should carefully consider the ambitious promotion of vehicle electrification in the heavy-duty segment.

Keeping these significant resource constraints in mind, it is recommended that the decarbonization of the HDV segment should rely on a broader mix of technologies, including fuel cell, biofuel, and natural gas vehicles. Among these alternatives, fuel cell vehicles fueled with hydrogen from renewable energy sources can simultaneously achieve zero carbon emissions and zero tail-pipe emissions and thus deserve more attentions. However, fuel cell vehicles also rely on critical resources, namely, platinum group metals (platinum, palladium and rhodium). These metals, characterized by high catalytic activities, are indispensable ingredients for fuel cells. The comparison of projected future global platinum group metal demand with resource endowment implies a less challenging future compared with the situation for lithium in a HDV electrification scenario[11]. It is therefore proposed that fuel cell vehicles should be prioritized for decarbonizing the HDV segment.

As demonstrated by scenario D3, reducing the required electric range of HDVs contributes to lowering lithium demand. Although it is not rational to promote mass electrification in the HDV segment, electric HDVs operated within specific contexts with relatively low-electric range and battery capacity requirements should be encouraged, such as mining trucks, port drayage trucks, urban delivery trucks, and transit buses. Dynamic charging and catenary charging infrastructures established along trunk routes can help further enable long-haul transport undertaken by low-electric range HDVs. However, reducing the electric range of HDVs is expected to lead to a significant increase in battery replacement and corresponding lithium outflow. This calls for attention to ensure a high end-of-life recycling rate for the battery replacement process.

The current level of battery durability is considered sufficient to eliminate the need for battery replacement for LDVs in most situations, but not for the high-mileage HDVs. As demonstrated by scenario D4, extending battery durability could effectively reduce the lithium demand from HDV battery replacement. This suggests that the HDV segment should have its own battery technology roadmap, with a high priority on improving battery durability. In particular, higher durability battery chemistries, such as lithium iron phosphate batteries offer possibilities for HDV applications. Battery durability-enhancing technologies, such as dry battery electrode technology, should be deployed as a priority. Furthermore, efforts should be made throughout the battery R&D, design, and manufacturing stages.

This study uses degradation to 80% of the initial battery capacity as the criterion for vehicle battery life end. Further potential could be extracted from these batteries through secondary use in fields with lower battery performance requirements, such as static energy storage systems for future smart grids[12]. When repurposing recycled vehicle batteries as energy storage systems, further battery degradation to 50% of the initial battery capacity is considered to be acceptable[13]. Existing studies have identified great opportunities for the secondary use of recycled vehicle batteries for grid-scale energy storage[13]. However, such an option should be carefully considered since several barriers exist for battery repurposing, including the high costs associated with the testing and reassembly of recycled battery cells, the lack of unified technological standards among different battery manufacturers, concerns for battery aging and related safety issues[12]. Further institutional and technological efforts are therefore needed to overcome these barriers.

This study is based upon the important assumption that batteries will maintain a consistent degree of reliance on lithium resources. The outcomes and conclusions would be completely different if next-generation lithium-free energy storage technologies achieve a breakthrough. For example, super capacitors, with no reliance on critical metals, offer the advantages of high charging rate, durability and power density. The major drawback that prevents the utilization of super capacitors in vehicles is their current low energy density. Another example is metal–air batteries, such as aluminum–air and magnesium–air batteries. These batteries promise high energy density at a low cost. However, they suffer from drawbacks of low durability and power density. While these technologies could be potential game changers, it should be kept in mind that future development of these technologies is highly uncertain. Integrated efforts from all stakeholders, including the government, industry, and research institutes, are needed to develop such innovative technologies so that pressures on global critical metal resources can be alleviated.

## Methods

**Model description.** The lithium inflow, outflow, and stock associated with PEVs are simulated by using the Transport Impact Model (TIM) developed by the China Automotive Energy Research Center at Tsinghua University[11,14]. TIM is a technology-rich, bottom-up approached model that is used to simulate the energy, environmental, and resource impacts from the global transport sector. The simulation covers 140 countries (or special regions) and five vehicle types (passenger vehicles and light-duty commercial vehicles are categorized into the LDV segment; medium-duty trucks, heavy-duty trucks, and heavy-duty buses are categorized into the HDV segment), with a one-year step length for the period of 2000–2100.

The total material inflow to the end-use vehicles is defined as gross demand. With gross demand subtracting the part that can be met by recycling, the rest of gross demand that has to be met by primary resources is defined as net demand. The net demand can be translated into mine production demand by considering resource efficiency of the entire production chain, which is about 90% for lithium-ion battery production[15,16].

**Scenarios.** The future lithium demand from vehicle electrification is affected by factors including vehicle sales growth, PEV market penetration, vehicle electric range, battery durability, battery lithium content, battery recycling, etc. This variety leads to a wide range of lithium demand possibilities in the future. Therefore, this study establishes four demand scenarios (D1, D2, D3, and D4) to reflect the major future possibilities and to examine the impacts from different factors, as summarized in Supplementary Table 3. Under scenario D1, the mass penetration of PEVs is restricted within the LDV segment, with no penetration in the HDV segment. Under scenario D2, PEVs gain an absolute techno-economic advantage over conventional internal combustion engine vehicles and other potential alternatives, achieving mass penetration in both the LDV and HDV segments. HDVs are assumed to have a normal electric range and unchanged battery durability. Under scenario D3, a reduced electric range of HDVs is further assumed. Under scenario D4, improved battery durability is further assumed. The assumptions behind the scenarios are explained as follows.

The vehicle electric range, which determines the single-vehicle battery capacity, is a critical factor affecting lithium demand. Conventional diesel HDVs normally offer a high range of up to over 1000 km because extending the vehicle range is simply a matter of enlarging the oil tank, which can be realized with a low cost. In contrast, the electric range of battery electric HDVs must be balanced between operation requirements and battery cost constraints. The electric ranges of the announced battery electric HDV models are significantly lower than the ranges of conventional vehicles, ranging from 100 to 800 km[2,4]. Furthermore, the need for electric range is substantially affected by the deployment of charging

infrastructures. A more intensive charging station network and the installation of dynamic charging or catenary charging facilities would contribute to reducing the need for electric range and the corresponding battery capacity. Therefore, two electric range cases, a normal electric range (500 km) and a reduced electric range (300 km), are established for HDVs to reflect future conditions resulting from charging infrastructure deployment.

Battery replacement in HDVs is potentially a major source of lithium demand. The number of battery replacements needed within a vehicle's lifetime is essentially determined by battery durability. Battery durability is commonly measured by the indicator of battery cycle life, which is defined as the number of full cycles a battery is able to deliver under specified operating conditions before failing to meet its specified end-of-life criteria (normally 80% of the battery's initial capacity)[17,18]. Current mainstream lithium nickel manganese cobalt oxide (NMC) battery technology endures a cycle life of up to 1000–2000 cycles[17]. When batteries are used in vehicles, battery life is influenced by the operating conditions, essentially temperature, depth of discharge and current[19]. Generally, higher temperature, greater depth of discharge, and high-rate charging/discharging damage battery life. While both favorable operating conditions (operator's willingness to extend battery life and avoid battery replacement costs; stable driving conditions on highways) and unfavorable operating conditions (a higher possibility of high depth of discharge for logistics efficiency; more extreme temperature conditions) exist for HDVs, the overall impact on battery life could be insignificant with a well-designed battery management system[20]. With this consideration, the battery life of HDVs is assumed to be at the same level as that of LDVs. Furthermore, enabled by advanced technologies, such as dry battery electrode technology, future battery durability could potentially improve, leading to reduced need for battery replacement. To reflect such influences, two battery durability cases are established, including one case in which battery durability remains unchanged (constant at 1000 cycles), and one in which the durability further improves (2000 cycles by 2030). The battery cycle life is translated into km-measured lifespan by multiplying the battery cycle life by the vehicle electric range.

**Common assumptions**. In addition to the above discussed factors, the following common underlying assumptions apply to all scenarios. The battery technology that supports the mass penetration of PEVs is the current generation of lithium-ion batteries, which have a consistent degree of reliance on lithium material; in terms of lithium recycling, a well-established recycling system is in place to ensure a high collection rate of end-of-life vehicle batteries. Furthermore, the lithium recovery technology for end-of-life batteries is well developed, resulting in an optimistic lithium recovery rate[10,21]. The recycling is assumed to be closed-loop recycling[22], namely, the recycled lithium compound reaches the quality for battery production. The open-loop recycling case, in which the recycled lithium cannot be used for

battery production, is not discussed in this study because open-loop recycling leads to a highly resource-unsustainable future even if only LDV electrification is considered[10]. The assumptions needed for the calculation of lithium demand are summarized in Table 1.

**Overall calculations**. Equations (1)–(6) present the major calculation flows embedded in the model to simulate the resource impacts. For a given vehicle type, country and year, the material inflow associated with vehicle manufacturing is calculated as the product of vehicle sales, battery capacity per vehicle, and material content per battery capacity. Similarly, the material outflow associated with vehicle scrappage is the accumulation of material content from vehicles that are scrapped in the given year; the material inflow associated with battery replacement is the accumulation of material content from vehicles that replace batteries in the given year; the material outflow associated with battery replacement equals to the material inflow; the material stock is the accumulation of material content from vehicles that are in use in the given year. The recycled material is calculated as the material outflow multiplied by the end-of-life recycling rate of the given material.

$$MIOE_{i,p,q,r} = SA_{i,p,q} \cdot BC_{i,p,q} \cdot MC_{i,p,q,r} \quad (1)$$

$$MOOE_{i,p,q,r} = \sum_{i_0 \leq j \leq i} SA_{j,p,q} \cdot \Delta SR^i_{j,p,q} \cdot BC_{j,p,q} \cdot MC_{j,p,q,r} \quad (2)$$

$$MIBR_{i,p,q,r} = \sum_{n} \sum_{i_0 \leq j \leq i} SA_{j,p,q} \cdot \Delta BR^i_{j,p,q,n} \cdot BC_{j,p,q} \cdot MC_{j,p,q,r} \quad (3)$$

$$MOBR_{i,p,q,r} = MIBR_{i,p,q,r} \quad (4)$$

$$MS_{i,p,q,r} = \sum_{i_0 \leq j \leq i} SA_{j,p,q} \cdot SR^i_{j,p,q} \cdot BC_{j,p,q} \cdot MC_{j,p,q,r} \quad (5)$$

$$SS_{i,p,q,r} = (MOOE_{i,p,q,r} + MOBR_{i,p,q,r}) \cdot RR_{i,p,q,r} \quad (6)$$

where

$MIOE_{i,p,q,r}$ is the inflow of type r material, in year i, by vehicle type p, in country q associated with vehicle manufacturing (g);

$MOOE_{i,p,q,r}$ is the outflow of type r material, in year i, by vehicle type p, in country q associated with vehicle scrappage (g);

$MIBR_{i,p,q,r}$ is the inflow of type r material, in year i, by vehicle type p, in country q associated with vehicle battery replacement (g);

$MOBR_{i,p,q,r}$ is the outflow of type r material, in year i, by vehicle type p, in country q associated with vehicle battery replacement (g);

**Table 1 Descriptions of model variables.**

| Variables | Descriptions | Details |
|---|---|---|
| Vehicle sales, scrappage and stock | The Shared Socio-economic Pathways (SSP) scenarios are employed as the basis for population and economic growth assumptions[23,24]. Vehicle sales, scrappage and stock projections are based on previous works by the authors[11,14,25]. Compared with other existing projections, the projected vehicle stock in this study shows high consistency[26–28]. | Supplementary Fig. 4 |
| Market penetration of PEVs | Case I (scenario D1): Mass electrification is restricted within the LDV segment. Case II (scenario D2, D3 and D4): Mass electrification is realized in both the LDV and HDV segments. For each case, four market penetration profiles are established distinguished by vehicle segment (LDV/HDV) and country development level (more developed countries/less developed countries). | Supplementary Fig. 5 |
| Electric range and battery capacity | Case I (Scenario D1 and D2): Electric range is assumed to be 500 km for battery electric HDVs, and 100 km for plug-in hybrid electric HDVs. Case II (Scenario D3 and D4): Benefiting from well-developed charging infrastructure, electric range of battery electric HDVs decreases to 300 km by 2030 and stays constant thereafter. For both cases, the electric range is assumed to be 300 km for battery electric LDVs, 60 km for plug-in hybrid electric LDVs. | Supplementary Figs. 6,7,8,9,10 |
| Battery durability | Case I (Scenario D1, D2 and D3): battery durability stays unchanged (constantly 1000 cycles). Case II (Scenario D4): battery durability finds further improvement (2,000 cycles by 2030). | Supplementary Figs. 11,12 |
| Lithium content | The lithium contents for different lithium-ion battery technologies are obtained from the BatPaC model developed by Argonne National Laboratory[29]. The average lithium content is 0.123 g/Wh. | |
| Recycling | The end-of-life recycling rate of lithium increases from the current level of basically 0% to 80% in 2030, reflecting both a well-established end-of-life battery collecting system and well-developed lithium recovery technologies[10]. | |

$MS_{i,p,q,r}$ is the stock of type r material, in year i, by vehicle type p, in country q (g);

$SS_{i,p,q,r}$ is the recycled material of type r material, in year i, by vehicle type p, in country q (g);

$RR_{i,p,q,r}$ is the end-of-life recycling rate of type r material, in year i, by vehicle type p, in country q (%);

$SA_{i,p,q}$ is the sales of type p vehicle, sold in year i, in country q;

$SR_{j,p,q}^{i}$ is the survival rate of type p vehicle, sold in year j, in country q, at the vehicle age of i-j (%);

$BR_{j,p,q,n}^{i}$ is the nth battery replacement rate of type p vehicle, sold in year j, in country q, at the vehicle age of i-j (%);

$BC_{i,p,q}$ is the average battery capacity of type p vehicle, sold in year i, in country q (Wh);

$MC_{i,p,q,r}$ is the material content (material volume/battery capacity) of type r material of type p vehicle, sold in year i, in country q (g/Wh).

**Battery capacity calculation**. The battery capacity is calculated by using Eqs. (7) and (8). On one hand, the battery capacity determines the total energy that can be used for driving; On the other hand, the battery capacity itself affects the battery weight, which further affects the energy consumption rate of PEVs. A larger battery capacity does not yield a proportionally larger electric range due to the increase in battery weight. The battery capacity can be obtained by solving the non-linear equations.

$$ER_{i,p,q} = \frac{BC_{i,p,q} \cdot \gamma}{EC_{i,p,q}} \quad (7)$$

$$EC_{i,p,q} = \alpha \cdot RF_{i,p,q} \cdot \frac{\left(BC_{i,p,q} \cdot ED_{i,p,q} + CW_{i,p,q}\right)^{\beta}}{PE_{i,p,q}} \quad (8)$$

where

$ER_{i,p,q}$ is the electric range of type p vehicle, sold in year i, in country q (km);

$EC_{i,p,q}$ is the energy consumption rate of type p vehicle, sold in year i, in country q (MJ/km), which is the function of vehicle weight, aero and rolling resistance, and vehicle powertrain efficiency;

$RF_{i,p,q}$ is the aero and rolling resistance factor of type p vehicle, sold in year i, in country q;

$ED_{i,p,q}$ is the battery energy density of type p vehicle, sold in year i, in country q (kg/MJ);

$CW_{i,p,q}$ is the vehicle curb weight (excluding battery weight) of type p vehicle, sold in year i, in country q (kg);

$PE_{i,p,q}$ is the powertrain energy efficiency of type p vehicle, sold in year i, in country q (%);

$\alpha$ and $\beta$ are the characteristics parameters, which reflect the rationale of vehicle energy consumption;

$\gamma$ is the percentage of battery energy that can be actually used out of the total battery capacity (%).

**Battery replacement calculation**. The battery cycle life is translated into km-measured lifespan by multiplying the battery cycle life with vehicle electric range, shown in Eq. (9).

$$BL_{i,p,q} = CL_{i,p,q} \cdot ER_{i,p,q} \quad (9)$$

where

$BL_{i,p,q}$ is the battery lifespan of type p vehicle, sold in year i, in country q (km);

$CL_{i,p,q}$ is the battery cycle life of type p vehicle, sold in year i, in country q (cycles).

It is assumed that when the vehicle travel distance reaches the battery lifespan, the vehicle owner considers battery replacement. Depending on the total vehicle lifespan and battery lifespan, battery replacement may not occur or occur many times. In reality, even if the battery lifespan is reached, the vehicle owner might not replace the battery if the remaining vehicle lifetime is low. To reflect this reality, the factor of vehicle owner Willingness-to-Replace (WtR) is incorporated. The WtR is defined as the share of vehicle owners that are willing to replace battery out of all vehicle owners that face battery replacement decision. The WtR is estimated through Eq. (10). It should be noted that battery replacement is only considered for BEVs. PHEVs are not considered due to their lower possibility of battery replacement and an overall negligible impact.

Given the vehicle lifespan and battery lifespan, after the vehicle travel distance reaches the nth battery lifespan: Case I: if the remaining vehicle lifetime is still higher than the battery lifespan (namely, a new battery will be fully utilized if battery replacement occurs), it is assumed that all vehicle owners are willing to replace the battery (WtR = 1). Case II: if the remaining vehicle lifetime is lower than the battery lifespan (namely, a new battery will only be partially utilized if battery replacement occurs), it is assumed that only part of the vehicle owners are willing to replace the battery. The WtR is measured by the ratio of remaining vehicle lifetime to battery lifespan. Two extreme situations under case II are: (a) if the remaining vehicle lifetime is zero, then no vehicle owner will replace the battery (WtR = 0); (b) if the remaining vehicle lifetime is equal to the battery lifespan, then all vehicle owners will replace the battery (WtR = 1, equivalent to case I).

$$WtR = \begin{cases} 1 & \text{if}: LS_{i,p,q} - n \cdot BL_{i,p,q} > BL_{i,p,q} \\[2ex] \frac{LS_{i,p,q} - n \cdot BL_{i,p,q}}{BL_{i,p,q}} & \text{if}: LS_{i,p,q} - n \cdot BL_{i,p,q} \le BL_{i,p,q} \end{cases} \quad (10)$$

where

WtR is the vehicle owner's willingness to replace the battery (%);

$LS_{i,p,q}$ is the vehicle lifespan of type p vehicle, sold in year i, in country q (km);

n is the number of battery lifespans that the vehicle travel distance have reached.

**Reporting summary**. Further information on research design is available in the Nature Research Reporting Summary linked to this article.

## Data availability

Data associated with this study can be accessed by browsing https://doi.org/10.6084/m9. figshare.10062512. The source data underlying Figs. 1, 2, Supplementary Figs. 1–12, Supplementary Table 2 are provided as a Source Data file.

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

## Acknowledgements

This study is sponsored by the National Natural Science Foundation of China (71774100, 71690241, 71403142, 71810107001), Young Elite Scientists Sponsorship Program by CAST (YESS20160140), State Key Laboratory of Automotive Safety and Energy (ZZ2019–023), Tsinghua-Rio Tinto Research Center for Resources Energy and Sustainable Development, and the big data project funded by Shanghai Jiao Tong University (SJTU-2019UGBD-03).

## Author contributions

H.H. conducted the modeling and calculations, wrote the paper; Y.G. coordinated the working project; J.E.T., F.L., K.C., X.S., Z.L., F.Z. contributed to the discussion of the results and implications and commented on the paper at all stages. Y.G. and H.H. led the revision work.

## Competing interests

The authors declare no competing interests.
