## [Peer Review File · Nature Communications]

Reviewers' comments:

Reviewer #1 (Remarks to the Author):

The paper claims that a complete conversion of light duty and heavy duty vehicles on a global scale to battery electric technology may lead to strain in lithium reserves. The paper provides interesting results, of importance to policy makers, researchers and industry, especially following recent trends in vehicle electrification.

However, the results and discussion section seem a little lackluster. A more in-depth discussion would strengthen the paper. Further, a number of sections are a little convoluted and some re-organizing may improve readability.

I would recommend acceptance given that the manuscript is thoroughly reviewed. Additional comments follow below.

Introduction:

1) The paper is well written, but it has a number of grammatical errors and sentences that do not make sense. e.g. lines 58-59 (gained progress), line 77, line 87 and line 89, to name a few. I recommend that the paper should be thoroughly proof read to address these errors. Further, the article has numerous uses of the determiner "the" in places where it is not necessary.

2) Paragraph structure could use some work. In the intro some of the paragraphs seem to have too many differing ideas. For example, in the second paragraph (starting in line 74), the first line of the paragraph talks about advantages of electrification of the HDV segment. But then at some point the authors start talking about drawbacks. Either the opening line of the paragraph should be modified to mention that there are both advantages and drawbacks, or drawbacks should be in a paragraph of their own.

3) Avoid the use of vernacular or meaningless words like "huge", "actually" or "quite a lot".

4) You mention in the intro that the lifetime of batteries is shorter than the lifetime of a truck and that multiple battery swaps would be necessary during a truck's lifetime. While I do not completely disagree with this claim, I believe it to be slightly exaggerated. Could you provide numbers and references to back up this claim?

5) Paragraph starting in line 92 – the authors mention resource impacts of LDV electrification and provide numerous references. What are these impacts? What type of resources and could this be quantified?

Methods section:

6) What are AERs? You must define all acronyms.

7) I believe some of the numbers used on vehicles in this section are exaggerated. A lifetime mileage of 1,500,000 km for a HD truck seems high to me, almost the worst case scenario. Do the authors have references to back up this claim? I would argue that the standard lifetime would be closer to 800,000 km – 1,000,000. See

https://www.theicct.org/sites/default/files/publications/Zero-emission-freight-trucks_ICCT-white-paper_26092017_vF.pdf, Keller et al, 2019, Energy and Telebian et al. 2018, Energy Policy.

Further, I would expect the lifetime of batteries for HD vehicles to be longer (in terms of km), rather than shorter than passenger vehicles. HD vehicles operating for profit would likely attempt to maximize their vehicle/ battery lifetime. Further, these vehicles tend to drive in highways, which I imagine would be beneficial for battery lifetime? Do the authors have sources to back up their claim?

8) This section is a little messy as well. The whole explanation on battery lifecycle and vehicle mileage belongs in the intro, not in the methods. Further, the authors buried the scenarios at the end of the paragraph where these vehicle mileages were explained. This made the paragraph messy and difficult to follow. Scenarios should be in a paragraph of their own.

9) What are your assumptions for the demand forecast? I know this is present in the supplemental

material but I would expect at least a couple of sentences here. If I required more in-depth info, then, and only then I would refer to supplemental material.

Results:

10) The authors briefly mention in the methods that the model includes medium and light duty commercial vehicles as well but I do not see those in the results. Depending on the jurisdiction, MD and LD demand can be significantly higher than HD. Why was this not considered?

11) The grammar in this section is rather poor. I would recommend a thorough review.

12) I agree with the main findings of the research piece that mass electrification of the LD + HD transport sector may put strain on global lithium demand, although as mention above, I believe the results may be slightly exaggerated. However, what I think lacks in this discussion is the view of the authors on what a more realistic pathway for de-carbonizing the HD sector is, if direct electrification is not feasible. Use of fuel cell vehicles? Renewable natural gas? bio-fuels? Please elaborate.

13) How do you expect other industries that impose a lithium demand to compare? i.e. consumer electronics, grid scale battery, etc. Would these all compete for the same resources?

14) In the results/ discussion I would have expected to see a further discussion on following topics:

- a. Fuel cell heavy duty vehicles and effects of mining of platinum
- b. Opportunities for use of older vehicle batteries for grid scale electricity storage
- c. breakthroughs in battery technology that may require lower lithium loading or replace it altogether.

Reviewer #2 (Remarks to the Author):

This paper do an study of heavy duty vehicle electrification on lithium ion resources and feasibility. Paper argues that it is not feasible to electrify the heavy duty vehicles due to limited lithuim ion resources and limited battery life from the developed model. Some of questions are

1. Battery technology is not just limited to lithium ion, there are other technologies which are being developed such as dry battery electrode technology can improve energy density.
2. Technologies such as dynamic charging of electric vehicle can reduce dependence of battery.
3. Ultracapacitor can be employed to reduce dependcy.
4. How the model is developed please explain.

Response to reviewer #1

Reviewer comments	Responses
The paper claims that a complete conversion of light duty and heavy duty vehicles on a global scale to battery electric technology may lead to strain in lithium reserves. The paper provides interesting results, of importance to policy makers, researchers and industry, especially following recent trends in vehicle electrification.	Dear reviewer: Thanks very much for your valuable comments. We carefully addressed each comment with our best efforts. The manuscript has been substantially improved with your help. Please find the responses to your comments below. Sincere, The authors
However, the results and discussion section seem a little lackluster. A more in-depth discussion would strengthen the paper. Further, a number of sections are a little convoluted and some re-organizing may improve readability.	We fully recognized the weaknesses in the results/discussion sections of the previous manuscript. A much more in-depth discussion is provided in the revised manuscript (see the response to your comment #14 for details). Furthermore, the sections of methods and results are re-organized to improve readability (see the responses to your comments #8, #9 for details).
I would recommend acceptance given that the manuscript is thoroughly reviewed. Additional comments follow below.	Thanks very much for your recognition! Please find the responses to your comments below.
Introduction: 1) The paper is well written, but it has a number of grammatical errors and sentences that do not make sense. e.g. lines 58-59 (gained progress), line 77, line 87 and line 89, to name a few. I recommend that the paper should be thoroughly proof read to address these errors. Further, the article has numerous uses of the determiner “the” in places where it is not necessary.	We are deeply sorry for the unsatisfying English presentation of the previous manuscript. Following your suggestion, one author of this paper, Professor James Tate from University of Leeds as a native English researcher, has re-edited the whole paper. Furthermore, we used the Nature Research editing service, under which the paper is polished by professional native English-speaking editors. The certification for the proof reading is attached for your consideration. We believe the writing quality of the revised manuscript has been significantly improved. Specifically, the errors pointed out in your comment have been carefully revised. The excessive uses of the determiner “the” have been carefully examined, with unnecessary uses removed.

2) Paragraph structure could use some work. In the intro some of the paragraphs seem to have too many differing ideas. For example, in the second paragraph (starting in line 74), the first line of the paragraph talks about advantages of electrification of the HDV segment. But then at some point the authors start talking about drawbacks. Either the opening line of the paragraph should be modified to mention that there are both advantages and drawbacks, or drawbacks should be in a paragraph of their own.	We are sorry for the unsatisfying paragraph structure in the previous manuscript, especially the example mentioned in your comment. To improve the paragraph structure, we carefully discussed each paragraph throughout the manuscript, making sure that each paragraph is self-consistent in its narrative and content. As for the example mentioned in your comment, we revised the first sentence of the paragraph as “Compared with conventional HDVs, electric HDVs show both advantages and drawbacks” so that it keeps consistent with the rest of the paragraph content.
3) Avoid the use of vernacular or meaningless words like “huge”, “actually” or “quite a lot”.	Following your recommendation, we carefully examined the uses of adjectives and adverbs, and removed those with no explicit meanings. Especially, the uses of “huge”, “actually” and “quite a lot” have all been removed from the text.
4) You mention in the intro that the lifetime of batteries is shorter than the lifetime of a truck and that multiple battery swaps would be necessary during a truck's lifetime. While I do not completely disagree with this claim, I believe it to be slightly exaggerated. Could you provide numbers and references to back up this claim?	The statement on the need for battery replacement from heavy-duty trucks can be verified partly based on field tests and partly based on theoretical analysis. Considering field tests, there are several on-going electric truck demonstration programs around the world. Numerous demonstration reports do mention that battery replacement is needed for electric trucks. For example, California Air Resource Board (CARB) implemented a statewide demonstration of forty-four (44) zero-emission battery electric and plug-in hybrid drayage trucks. In the assessment for cost of ownership, CARB (2019) assumed that “no battery replacement for passenger van or delivery van, assumed to be necessary for regional haul tractor” (page 27, access: https://ww2.arb.ca.gov/sites/default/files/2019-02/190225actpres.pdf). San Pedro Bay Ports (2019) reported in the evaluation report of the electric truck demonstration program that “A battery-electric truck operator should anticipate that the maximum range of the truck could degrade to 80% of its original range over the course of its service life” (page 63, access:

	http://www.cleanairactionplan.org/documents/raft-dravage-truck-feasibility-assesment.pdf. However, due to the fact that these demonstration programs are all at their early stages, the accurate survey of the times of battery replacements needed is not available yet. Considering theoretical analysis, we had correspondence with one of the global recognized research laboratories in the field of battery aging mechanism - the professor Ouyang team at Tsinghua University (https://www.journals.elsevier.com/etransportation/editorial-board/minggao-ouyang). While they do agree that battery replacement is necessary for electric trucks, they raised the same concern as your comment that a 500-cycle battery life (50% of battery life on passenger vehicles) resulting in 3-4 times of battery replacements could be a bit exaggerated. Based on careful discussions, the experts agreed that 1000 cycles (same as battery life on passenger vehicles) could be the reasonable assumption (see the response to your comment #7 for details). This revised assumption leads to 1-2 times of battery replacements during a truck's lifetime, which is generally in line with the field test evidence.
5) Paragraph starting in line 92 – the authors mention resource impacts of LDV electrification and provide numerous references. What are these impacts? What type of resources and could this be quantified?	For ease of understanding and comparison, we added a table summarizing the resource impacts revealed by the cited literatures (see Supplementary Table 1 for details). The twenty-five literatures summarized in the table quantified the resource impacts of lithium, cobalt and nickel (part of the metals in some cases).
Methods section: 6) What is AERs? You must define all acronyms.	(Please note that the Methods section is moved to the end of the paper as per the Nature Communication paper formatting requirement) AER is the abbreviation for All-Electric Range, which is defined as the range that can be sustained by vehicle on-board electricity

	storage. The definition and abbreviation for AER were provided in the introduction section where they appear for the first time. It should be noted that after careful discussion, we replaced “AER” with the term “vehicle electric range” throughout the paper because the latter term is more commonly used and easily understood.
7) I believe some of the numbers used on vehicles in this section are exaggerated. A lifetime mileage of 1,500,000 km for a HD truck seems high to me, almost the worst case scenario. Do the authors have references to back up this claim? I would argue that the standard lifetime would be closer to 800,000 km – 1,000,000. See https://www.theicct.org/sites/default/files/publications/Zero-emission-freight-trucks_ICCT-white-paper_26092017_vF.pdf, Keller et al, 2019, Energy and Telebian et al. 2018, Energy Policy. Further, I would expect the lifetime of batteries for HD vehicles to be longer (in terms of km), rather than shorter than passenger vehicles. HD vehicles operating for profit would likely attempt to maximize their vehicle/ battery lifetime. Further, these vehicles tend to drive in highways, which I imagine would be beneficial for battery lifetime? Do the authors have sources to back up their claim?	Regarding the lifetime mileage assumption, we carefully examined the three literatures provided in your comment. The ICCT report assumed ten-year mileage of long-haul heavy-duty vehicles to be 1,390,490 km, 834,294 km and 1,011,563 km for the U.S., EU28 and China, respectively. While the U.S. estimation is based on solid source, the EU28 and China are assumed to be 40% and 27% lower than the U.S. with no specific explanations. The Keller et al. and Talebian et al. studies are using the same lifetime mileage assumption of 900,000 km, which is based on the Canada Alberta context. Based on this, we further searched more literatures in this field. In the annually updated trucking operational cost report published by the American Transport Research Institute (ATRI), they provided survey-based estimations for lifetime mileage of tractor trailers in the U.S. context, which is about 1,200,000 km (https://truckingresearch.org/atri-research/operational-costs-of-trucking/). This ATRI report is widely cited and is the most reliable estimation for the U.S. to the best of our knowledge. Regarding EU, another ICCT report provides very convincing estimations for lifetime mileage, which is about 1,000,000 km for long-haul tractor trailers (https://theicct.org/publications/cost-effectiveness-of-fuel-efficiency-tech-tractor-trailers). Estimations for other countries are quite scarce. Based on these investigations, we revised the lifetime mileage of the tractor trailers to be 1,200,000 for the U.S. and 1,000,000 for the EU. Due to the lack of data, other countries are

	assumed to follow the EU pattern. Correspondingly, the need for battery replacement from heavy-duty vehicles is reduced, resulting in a decline in the battery replacement related lithium demand. Regarding the battery lifetime, as the electric truck demonstration programs are on the early stages, we tried our best but could not find field test-based battery lifetime estimations for electric trucks. Theoretically, The factors determining battery lifetime are mainly temperature, depth of discharge, and current. As mentioned in your comment, we fully agree that there are factors that benefit battery lifetime on electric trucks, including the operator's consideration to avoid battery replacement cost, the stable driving condition on highways. Meanwhile, there are also factors that tend to harm battery lifetime, including higher possibility of high depth of discharge (for logistics efficiency), tougher temperature condition, high-rate discharging under heavy load and uphill. As mentioned above, we discussed this issue with the professional battery aging mechanism research team. Based on the discussion, we agreed that while many factors affecting battery aging exist, most of them can be addressed with well-designed battery management system, thus the difference of battery cycle life between LDV and HDV should be insignificant. Based on this consideration, we assumed the same battery cycle life for LDV and HDV (compared with the previous assumption that battery cycle life of HDV is 50% lower than LDV). The need for battery replacement and the associated lithium demand from HDVs are further reduced.
8) This section is a little messy as well. The whole explanation on battery lifecycle and vehicle mileage belongs in the intro, not in the methods. Further, the authors buried the scenarios at the end of the paragraph where these vehicle mileages were explained. This	The whole methods section is re-organized in the revised manuscript, with a focus on describing the scenarios and assumptions. Specifically, by following the recommendation from reviewer #2, we re-arranged the scenario narratives and established four scenarios

made the paragraph messy and difficult to follow. Scenarios should be in a paragraph of their own.	(D1/D2/D3/D4) distinguished by PEV market penetration, vehicle electric range and battery durability. The scenario narratives are dedicatedly described in the second part of the section. The rest of the section provides the detailed assumptions (see the Methods section for details). Regarding the descriptions of vehicle life and battery life, the description of vehicle life is moved to the introduction section as background information. The description of battery life is kept in the methods section because it is one essential element for the scenario narratives.
9) What are your assumptions for the demand forecast? I know this is present in the supplemental material but I would expect at least a couple of sentences here. If I required more in-depth info, then, and only then I would refer to supplemental material.	We fully recognized the issue pointed out in your comment that the assumptions in the paper are not provided in an efficient way, making them difficult to follow. To address this issue, we added a table summarizing major assumptions. The table provides (1) direct access to the most essential information, and (2) index of the in-depth information in the Supplementary Information. With this table added, the assumptions can be much more easily followed (see Table 1 for details).
Results: 10) The authors briefly mention in the methods that the model includes medium and light duty commercial vehicles as well but I do not see those in the results. Depending on the jurisdiction, MD and LD demand can be significantly higher than HD. Why was this not considered?	In this model, vehicles are classified into five vehicle types: (1) passenger vehicles, (2) light-duty commercial vehicles, (3) medium-duty trucks, (4) heavy-duty trucks, and (5) heavy-duty buses. To avoid complexity in the analysis, the five vehicle types are further categorized into the Light-duty vehicle segment (LDV, including vehicle types (1)&(2)) and the heavy-duty vehicle segment (HDV, including vehicles types (3)(4)&(5)). All the results in the main text are provided with the LDV/HDV segmentation. The results with further breakdown into the five vehicle types are provided in the Supplementary Information. Back to your comment, the light-duty and medium-duty commercial vehicles are covered in the results in the main text, but not provided as single entries. The light-duty commercial

	vehicles are part of the LDV entry, while the medium-duty commercial vehicles are part of the HDV entry. The explicit results for light-duty and medium-duty commercial vehicles can be found in the Supplementary Information (see Supplementary Figs. 1-3, Supplementary Table 2 for details).
11) The grammar in this section is rather poor. I would recommend a thorough review.	We are deeply sorry for the unsatisfying English presentation of the previous manuscript. To improve the grammar and wording, one author of this paper, Professor James Tate from University of Leeds as a native English researcher, has re-edited the whole paper. We also used the Nature Research Editing service, under which the paper is polished by professional native English-speaking editors. We believe the writing quality of the revised manuscript has been significantly improved.
12) I agree with the main findings of the research piece that mass electrification of the LD + HD transport sector may put strain on global lithium demand, although as mention above, I believe the results may be slightly exaggerated. However, what I think lacks in this discussion is the view of the authors on what a more realistic pathway for de-carbonizing the HD sector is, if direct electrification is not feasible. Use of fuel cell vehicles? Renewable natural gas? bio-fuels? Please elaborate.	Following your suggestion, we added the discussion on the realistic pathway for HDV segment decarbonization in the revised manuscript. From our perspective, the fuel cell vehicles should be the prioritized pathway (see the second paragraph in the Discussion section for details). [Quotation of added discussion] With these significant resource constraints in mind, it is recommended that the decarbonization of the HDV segment should rely on a broader mix of technologies including fuel cell, biofuel, and natural gas vehicles. Among these alternatives, fuel cell vehicles fueled with hydrogen from renewable energy sources are the only technology that can simultaneously realize zero carbon emissions and tailpipe emissions and thus bear the greatest expectations. Fuel cell vehicles however also rely on critical resources, namely, platinum group metals (platinum, palladium and rhodium). These metals, characterized by high catalytic activities, are indispensable ingredients for fuel cells. The comparison of projected future global platinum

	group metal demand with resource endowment implies a less challenging future compared with the situation for lithium in a HDV electrification scenario ¹¹. It is therefore proposed that fuel cell vehicles should be the prioritized solution for decarbonizing the HDV segment.
13) How do you expect other industries that impose a lithium demand to compare? i.e. consumer electronics, grid scale battery, etc. Would these all compete for the same resources?	As mentioned in your comment, one weakness of the previous manuscript is that the impact from industries other than electric vehicles was not considered. To address this issue, we added dedicated discussions on such unconsidered impact in the revised manuscript. To enable the discussion, the lithium demand from other industries was cited from one recent representative study from Ziemann et al. (2018). It turns out that the demand from other industries is comparable to about one quarter of the demand from electric vehicles. By considering such demand, the resource space for HDV electrification is further limited. This further strengthens the basis for our argument (see the last paragraph of the Results section for details). [Quotation of discussion] It should be noted that the estimations do not cover lithium demand from sectors other than PEVs. Existing studies show that lithium demand from other sectors is expected to grow in a relatively mild pattern in the coming decades. With an annual assumed growth rate of 5%, the gross demand is expected to reach around 0.2 mt by 2050 ¹⁰, equivalent to approximately one-quarter of the expected lithium gross demand from PEVs (under scenario D2). This modest but not insignificant additional demand adds further pressures of resources to support HDV electrification.
14) In the results/ discussion I would have expected to see a further discussion on following topics: a. Fuel cell heavy duty vehicles and effects of mining of platinum	The three topics proposed in your comment provide important implications and add great values to this paper. Following your suggestion, we added dedicated discussions on the three topics in the revised manuscript. The added

b. Opportunities for use of older vehicle batteries for grid scale electricity storage c. breakthroughs in battery technology that may require lower lithium loading or replace it altogether.	discussions are quoted below (see paragraphs 2, 5, 6 in the Discussion section for details). a. Fuel cell heavy duty vehicles and effects of mining of platinum: [Quotation of added discussion] With these significant resource constraints in mind, it is recommended that the decarbonization of the HDV segment should rely on a broader mix of technologies including fuel cell, biofuel, and natural gas vehicles. Among these alternatives, fuel cell vehicles fueled with hydrogen from renewable energy sources are the only technology that can simultaneously realize zero carbon emissions and tailpipe emissions and thus bear the greatest expectations. Fuel cell vehicles however also rely on critical resources, namely, platinum group metals (platinum, palladium and rhodium). These metals, characterized by high catalytic activities, are indispensable ingredients for fuel cells. The comparison of projected future global platinum group metal demand with resource endowment implies a less challenging future compared with the situation for lithium in a HDV electrification scenario ¹¹. It is therefore proposed that fuel cell vehicles should be the prioritized solution for decarbonizing the HDV segment. b. Opportunities for use of older vehicle batteries for grid scale electricity storage: [Quotation of added discussion] This study uses degradation to 80% of the initial battery capacity as the criterion for vehicle battery life end. Further potential could be extracted from these batteries through secondary use in fields with lower battery performance requirements, such as static energy storage systems for future smart grids ¹². When repurposing recycled vehicle batteries as energy storage systems, further battery degradation to 50% of the initial
--	--

battery capacity is considered to be acceptable¹³. Existing studies have identified great opportunities for the secondary use of recycled vehicle batteries for grid-scale energy storage¹³. However, such an option should be carefully considered since several barriers exist for battery repurposing, including the high costs associated with the testing and reassembly of recycled battery cells, the lack of unified technological standards among different battery manufacturers, concerns for battery aging and related safety issues¹². Further institutional and technological efforts are therefore needed to overcome these barriers.

c. breakthroughs in battery technology that may require lower lithium loading or replace it altogether:

[Quotation of added discussion] This study is based upon the important assumption that batteries will maintain a consistent degree of reliance on lithium resources. The outcomes and conclusions would be completely different if next-generation lithium-free energy storage technologies achieve a breakthrough. For example, super capacitors, with no reliance on rare metals, offer the advantages of high charging rate, durability and power density. The major drawback that prevents the utilization of super capacitors in vehicles is their current low energy density. Another example is metal-air batteries, such as aluminum-air and magnesium-air batteries. These batteries promise high energy density at a low cost. However, they suffer from drawbacks of low durability and power density. While these technologies could be potential “game changers”, it should be kept in mind that the future development of these technologies is highly uncertain. Integrated efforts from all stakeholders, including the government, industry, and research institutes, are needed to develop such alternative technologies so that

	pressures on global resources can be alleviated.
--	--

Response to reviewer #2

Reviewer comments	Responses
This paper do an study of heavy duty vehicle electrification on lithium ion resources and feasibility. Paper argues that it is not feasible to electrify the heavy duty vehicles due to limited lithuim ion resources and limited battery life from the developed model. Some of questions are	Dear reviewer: Thanks very much for your valuable comments. We carefully addressed each comment with our best effort. The manuscript has been substantially improved with your help. Please find the responses to your comments below. Sincere, The authors
1. Battery technology is not just limited to lithium ion, there are other technologies which are being developed such as dry battery electrode technology can improve energy density.	As mentioned in your comment, the dry battery electrode technology is a very important technology that could potentially change the battery industry. We further investigated this technology and discussed its possible influence on our analysis. The dry battery electrode technology, mainly pioneered by Maxwell, is a liquid-free electrode fabrication technology that enables higher energy density, lower cost and longer durability for batteries. The major influence of this technology on our study is the extension of battery durability. This influence can be reflected in the narrative of scenario D4, under which battery durability is assumed to be doubled by 2030, reducing the need for battery replacement and associated lithium demand. Based on the results of scenario D4, we discussed the role dry battery electrode technology could play in reducing lithium demand from electric vehicles (see the descriptions and discussions on scenario D4 for details). [Quotation of scenario D4 establishment] ... Furthermore, enabled by advanced technologies such as dry battery electrode technology, future battery durability could potentially improve, reducing the need for battery replacement. To reflect such influences, two battery durability cases are established: one case in which battery

	durability remains unchanged (constant at 1,000 cycles), and one in which the durability further improves (2,000 cycles by 2030). [Quotation of discussions on scenario D4] The current level of battery durability is considered sufficient to eliminate the need for battery replacement for LDVs in most situations, but not for the high-mileage HDVs. As demonstrated by scenario D4, extending battery durability could effectively reduce the lithium demand from HDV battery replacement. This suggests that the HDV segment should have its own battery technology roadmap, with a high priority on improving battery durability. In particular, higher durability battery chemistries such as lithium iron phosphate batteries offer possibilities for HDV applications. Battery durability-enhancing technologies, such as dry battery electrode technology, should be deployed as a priority. Furthermore, efforts should be made throughout the battery R&D, design, and manufacturing stages.
2. Technologies such as dynamic charging of electric vehicle can reduce dependence of battery.	Dynamic charging and other advanced charging infrastructures potentially reduce the dependence on vehicle electric range and battery capacity. To reflect the impact from such an important factor, we established a new scenario (scenario D3), under which the dynamic charging and other advanced charging infrastructures are assumed to be better deployed, leading to lower vehicle electric range and battery capacity of HDVs. Based on this scenario, the impacts from advanced charging technologies are discussed (see the descriptions and discussions on scenario D3 for details). [Quotation of scenario D3 establishment] ... Furthermore, the need for electric range is substantially affected by the deployment of charging infrastructures. A more intensive charging station network and the installation of dynamic charging or catenary charging

facilities would contribute to reducing the need for electric range and the corresponding battery capacity. Therefore, two electric range cases, a normal electric range (500 km) and a reduced electric range (300 km), are established for HDVs to reflect future conditions resulting from charging infrastructure deployment.

[*Quotation of scenario results*] With an assumed reduction in the HDV electric range (scenario D3), the lithium gross demand from HDVs changes significantly. On the one hand, the lithium gross demand from HDV manufacturing decreases from 21.5 mt to 13.1 mt. This decrease reflects the impact from reduced single-vehicle battery capacity. On the other hand, the lithium gross demand from HDV battery replacement increases from 14.6 mt to 20.5 mt. The underlying reason behind this increase is that reducing the electric range of HDVs causes a reduction in the km-measured battery lifespan (obtained by multiplying the battery cycle life and vehicle electric range), which ultimately causes more frequent battery replacements. Overall, scenario D3 is estimated to lead to 81.9 mt of gross demand and 31.6 mt of net demand, 3% and 7% lower than the values under scenario D2, respectively.

[*Quotation of discussions*] As demonstrated by scenario D3, reducing the required electric range of HDVs contributes to lowering lithium demand. Despite the irrationality of mass electrification in the HDV segment, electric HDVs operated within specific contexts with relatively low electric range and battery capacity requirements should be encouraged, such as mining trucks, port drayage trucks, urban delivery trucks and transit buses. Furthermore, dynamic charging and catenary charging infrastructures established along trunk routes offer possibilities to enable long-haul transport undertaken by low-electric

	range HDVs. It should be noted that reducing the electric range of HDVs is expected to lead to a significant increase in requirements to replace batteries and corresponding lithium scrappage. This calls for attention to ensure a high recycling rate for the battery replacement process.
3. Ultracapacitor can be employed to reduce dependency.	It is true that ultra-capacitors can be employed as a potential alternative to lithium-ion batteries. To cover such an important topic, we extended the discussion to the potential impacts from ultra-capacitors and other rare-metal-free energy storage technologies (see the last paragraph of the Discussion section for details). [Quotation of the discussion on super capacitor] This study is based upon the important assumption that batteries will maintain a consistent degree of reliance on lithium resources. The outcomes and conclusions would be completely different if next-generation lithium-free energy storage technologies achieve a breakthrough. For example, super capacitors, with no reliance on rare metals, offer the advantages of high charging rate, durability and power density. The major drawback that prevents the utilization of super capacitors in vehicles is their current low energy density. Another example is metal-air batteries, such as aluminum-air and magnesium-air batteries. These batteries promise high energy density at a low cost. However, they suffer from drawbacks of low durability and power density. While these technologies could be potential “game changers”, it should be kept in mind that the future development of these technologies is highly uncertain. Integrated efforts from all stakeholders, including the government, industry, and research institutes, are needed to develop such alternative technologies so that pressures on global resources can be alleviated.
4. How the model is developed please explain.	We recognized the issue that the model was not

	well explained in the previous manuscript. The difficulty we encountered is that due to space limit, only the most essential information of the model can be provided in the main text. The massive in-depth model descriptions have to be provided in the Supplementary Information as an attachment, including the calculation flows and detailed assumptions (see the Supplementary Methods for details). To address this issue, we created a new table in the main text summarizing the essential model elements. The table serves as a guideline for understanding how the model is developed. It provides direct access to the most essential model descriptions, and an index of the in-depth information in the Supplementary Information. With this table added, the model can be much more easily followed (see Table 1 for details). Besides, to better explain the model, we re-organized the methods section. The model framework is firstly described, followed by the descriptions of scenarios and major assumptions.
--	--

REVIEWERS' COMMENTS:

Reviewer #1 (Remarks to the Author):

They did a really good job at addressing all my comments. I am (more than) satisfied.

Reviewer #3 (Remarks to the Author):

This manuscript highlights the sustainability (or lack there of) of the Lithium resources to supply the needs of an electrified transportation which includes the heavy duty vehicles. The overall observations are alarming and can be very useful for all stakeholders and policy makers. As one of the alternative solutions(suggestions) dynamic charging of electric vehicles could have been addressed. This would have reduced the amount of onboard battery storage system. Furthermore, monitoring of state of health in the battery storage system and adequate charging can prolong their active useful time. This, in turn, can impact the findings of the report. This reviewer finds the article very useful and clear in its message.

Response to reviewer #1

Reviewer comments	Responses
They did a really good job at addressing all my comments. I am (more than) satisfied.	Dear reviewer: Thanks very much for your nice comment. The paper would not have been so much improved without your valuable input! Sincere, The authors

Response to reviewer #3

Reviewer comments	Responses
This manuscript highlights the sustainability (or lack there of) of the Lithium resources to supply the needs of an electrified transportation which includes the heavy duty vehicles. The overall observations are alarming and can be very useful for all stakeholders and policy makers.	Dear reviewer: Thanks very much for your nice comment. Please find the response to your comment below. Sincere, The authors
As one of the alternative solutions(suggestions) dynamic charging of electric vehicles could have been addressed. This would have reduced the amount of onboard battery storage system. Furthermore, monitoring of state of health in the battery storage system and adequate charging can prolong their active useful time. This, in turn, can impact the findings of the report.	It is very important that dynamic charging technology could potentially contribute to reducing the amount of onboard battery storage system. Such impact is characterized by Scenario D3 in the paper, under which a reduced electric range is assumed for HDVs resulting from the deployment of advanced charging infrastructures, in particular dynamic charging. The implications from Scenario D3 are discussed in details in the paper (see the Methods section paragraph #4 and the Discussion section paragraph # 3 for details). [Quotation of scenario D3 description]...the need for electric range is substantially affected by the deployment of charging infrastructures. A more intensive charging station network and the installation of dynamic charging or catenary charging facilities would contribute to reducing the need for electric range and the corresponding battery capacity. Therefore, two electric range cases, a normal electric range (500 km) and a reduced electric range (300 km), are established for HDVs to reflect future conditions resulting from charging infrastructure deployment... [Quotation of scenario D3 discussion]...As demonstrated by scenario D3, reducing the required electric range of HDVs contributes to lowering lithium demand...Furthermore, dynamic charging and catenary charging

	infrastructures established along trunk routes offer possibilities to enable long-haul transport undertaken by low-electric range HDVs... As mentioned in your comment, monitoring of state of health in the battery storage system and adequate charging can indeed enhance battery life and reduce the need for battery replacement. The impact of longer battery life is characterized by Scenario D4, under which battery durability is assumed to be twice of the baseline level (2,000 cycles by 2030). The results indicate that extending battery durability could effectively reduce the lithium demand from HDV battery replacement (see the Methods section paragraph #5 and the Discussion section paragraph # 4 for details). [Quotation of scenario D4 description]...When batteries are used in vehicles, battery life is influenced by the operating conditions, essentially temperature, depth of discharge and current. Generally, higher temperature, greater depth of discharge, and high-rate charging/discharging damage battery life...To reflect such influences, two battery durability cases are established: one case in which battery durability remains unchanged (constant at 1,000 cycles), and one in which the durability further improves (2,000 cycles by 2030)... [Quotation of scenario D4 discussion]...As demonstrated by scenario D4, extending battery durability could effectively reduce the lithium demand from HDV battery replacement. This suggests that the HDV segment should have its own battery technology roadmap, with a high priority on improving battery durability...
This reviewer finds the article very useful and clear in its message.	Thanks very much for your nice comment!